# Estrogen Receptors: Therapeutic Perspectives for the Treatment of Cardiac Dysfunction after Myocardial Infarction

**DOI:** 10.3390/ijms22020525

**Published:** 2021-01-07

**Authors:** Jaqueline S. da Silva, Tadeu L. Montagnoli, Bruna S. Rocha, Matheus L. C. A. Tacco, Sophia C. P. Marinho, Gisele Zapata-Sudo

**Affiliations:** 1Programa de Pesquisa em Desenvolvimento de Fármacos, Instituto de Ciências Biomédicas, Universidade Federal do Rio de Janeiro, Rio de Janeiro 21941-902, Brazil; ssjck@hotmail.com (J.S.d.S.); tmontagnoli@gmail.com (T.L.M.); brunacruzdemalta@gmail.com (B.S.R.); linc.matheus@gmail.com (M.L.C.A.T.); sophiamarinhocp@gmail.com (S.C.P.M.); 2Instituto de Cardiologia Edson Saad, Faculdade de Medicina, Universidade Federal do Rio de Janeiro, Rio de Janeiro 21941-902, Brazil

**Keywords:** estrogen receptor, myocardial infarction, ischemia-reperfusion, cardiovascular disease, cardiac dysfunction

## Abstract

Estrogen receptors (ER) mediate functions beyond their endocrine roles, as modulation of cardiovascular, renal, and immune systems through anti-inflammatory and anti-apoptotic effects, preventing necrosis of cardiomyocytes and endothelial cells, and attenuating cardiac hypertrophy. Estradiol (E2) prevents cardiac dysfunction, increases nitric oxide synthesis, and reduces the proliferation of vascular cells, yielding protective effects, regardless of gender. Such actions are mediated by ER (ER-alpha (ERα), ER-beta (ERβ), or G protein-coupled ER (GPER)) through genomic or non-genomic pathways, which regulate cardiovascular function and prevent tissue remodeling. Despite the extensive knowledge on the cardioprotective effects of estrogen, clinical studies conducted on myocardial infarction (MI) and cardiovascular diseases still include favorable and unfavorable profiles. The purpose of this review is to provide up-to-date information regarding molecular, preclinical, and clinical aspects of cardiovascular E2 effects and ER modulation as a potential therapeutic target for the treatment of MI-induced cardiac dysfunction.

## 1. Introduction

Cardiovascular diseases (CVD) are the leading cause of death worldwide and still represent a crescent burden [1], as they accounted for nearly one-third of global deaths in 2016 and are expected to cause a further three million deaths by 2030, according to the World Health Organization. Myocardial infarction (MI) is a serious manifestation of coronary heart disease and represents a substantial global health problem, affecting more than seven million people in the world each year [2] and causing more than a third of deaths in developed nations annually [3]. Atherosclerotic cardiovascular disease is responsible for numerous cardiovascular events, including MI, and remains the leading cause of morbidity and mortality worldwide [4]. Moreover, ischemic heart disease stands as the most common isolated cause of death globally, and its frequency is increasing globally [5].

Although heart failure (HF) is the leading cause of death regardless of sex, women tend to develop HF later and have a better prognosis [6]. MI remains the most common cause of HF, despite notable advances in the treatment of coronary artery disease (CAD) and MI over the past two decades [7]. Thus, early treatment of HF induced by MI is an important clinical strategy for a better prognosis. Investigation on the cardiovascular role of estrogen receptors (ER) has recently evidenced them as potential pharmacological targets for cardioprotection, and their modulation is currently under study for the development of new therapeutic strategies for the treatment of CVD. Until now, three subtypes of ER are known, two of which are nuclear receptors, ER-alpha (ERα) and ER-beta (ERβ), and a third G protein-coupled ER (GPER). Over the years, their presence and actions have been demonstrated in different tissues and organs apart from the reproductive system, as well as the involvement of estrogen deficiency in pathological processes, such as immune cell activation, endothelial dysfunction, atherosclerosis, and as a risk factor for the development of CVD [8]. Activation of ER promotes genomic and non-genomic mechanisms, which result in anti-inflammatory and anti-apoptotic properties, in addition to cytoprotective effects on cardiac and endothelial cells and attenuate pathological cardiac hypertrophy [9]. For this reason, those receptors are now regarded as potential targets for the treatment of MI-induced HF.

## 2. Estrogen Receptors and Distribution in the Cardiovascular System

Classically, estrogen exerts its actions by activating ERα and ERβ, which are encoded by the *ESR1* and *ESR2* genes, respectively [10,11]. These receptors bind 17β-estradiol (E2) with similar affinity in their C-terminal ligand-binding domains, which share 60% sequence homology between subtypes and are responsible for receptor activation and dimerization. Transcriptional activity is guided by nearly identical DNA-binding domains and two additional activation function domains [11,12,13]. In 1996, the orphan G protein-coupled receptor GPR30 was first identified, and subsequent studies demonstrated its binding to E2 with high affinity, although less than nuclear ER, leading to its current designation as GPER [14].

All subtypes of ER have a wide distribution pattern in the body and control important physiological functions, including reproductive, cardiovascular, muscular, and in the central nervous system. Both ERα and ERβ are expressed in the cardiovascular system [15], particularly in cardiomyocytes, cardiac fibroblasts (CF), endothelium and epithelial cells in both male and female rats [16], as well as vascular smooth muscle cells (VSMC) of humans [16]. Despite their predominant cytoplasmic localization and genomic effects, ERα and ERβ can also promote rapid non-genomic cell signaling when embedded in the cell membrane. Detection of ERα and ERβ in mitochondria [16,17] and ERβ in adipose tissue and immune system [15] has also been reported.

Several studies identified *GPER* mRNA in the kidney and heart [18], including endothelial cells from cardiac and renal arterioles, and in smooth muscle cells [19]. GPER shows a broad distribution in the human heart, being expressed in both ventricles and atria, as well as in the atrioventricular sinus and aorta [20]. Moreover, in male mice ventricles, *GPER* mRNA is three- and seventeen-times more abundant than *ESR1* and *ESR2* mRNA, respectively [21]. It also displays a phenotype-dependent subcellular location, as its presence in endoplasmic reticulum membranes has been demonstrated [14] as well as in the plasma membrane. In cardiomyocytes, GPER is located in the cell membrane, cytoplasm [20,22] and mitochondria [22]. In addition, it is also found in VSMC cytoplasm and in the endoplasmic reticulum of endothelial cells [23].

Therefore, as the different ER subtypes show a wide distribution in most cell types of the cardiovascular system, their actions have been extensively investigated for the treatment of CVD, including MI, which will be further explored in the following sessions.

## 3. General Mechanism of Action of ER

Estrogen binding to ERα or ERβ in their monomeric form promotes homo- or hetero-dimerization, nuclear translocation, and association to co-regulators of gene transcription (canonical signaling) [24,25] and also may indirectly regulate gene expression by interactions with c-Jun and c-Fos [25], as demonstrated by the reduction in c-Jun-mediated gene transcription by E2 [26]. However, cell membrane ERα and ERβ [27] trigger rapid phosphorylation cascades through membrane-associated proteins. ERα enhances endothelial nitric oxide synthase (eNOS) activity in endothelial cells by stimulation of phosphatidylinositol 3-kinase (PI3K), while endothelial ERβ in caveolae also leads to NO production enhancement [10]. E2 promotes vasodilation by genomic pathways, increasing *NOS3* transcription and exhibits anti-mitogenic effects in smooth muscle cells by inhibition of mitogen-activated protein kinase (MAPK) [28], whereas, in endothelial cells, it stimulates eNOS and vascular endothelial growth factor (VEGF), both involved with cell proliferation [9,28]. Moreover, ER activation of PI3K/Akt pathway also inhibits the pro-apoptotic effects of transcription factors belonging to the forkhead family (FKHR) while stimulates gene transcription of anti-apoptotic proteins through nuclear factor kappa B (NF-κB) [29]. ERβ activation of PI3K and protein kinase A (PKA) also blocks the development of cardiac hypertrophy and fibrosis [26,27].

Activation of GPER produces vasorelaxation through an increase in 3′,5′-cyclic adenosine monophosphate (cAMP) [23,30], caused by the release of endothelial nitric oxide (NO) [30,31] and the inhibition of the vasoconstrictor activity of endothelial-derived prostanoids [32]. GPER also inhibits endothelial cell and VSMC proliferation [33]. In VSMC, GPER seems to be involved in extracellular signal-regulated kinase (ERK) phosphorylation [34] and activation of c-Fos by either ERK or PI3K pathways [35]. The proliferation of rat CF is limited by GPER through suppressing cell cycle proteins cyclin B1 and cyclin-dependent kinase 1 (CDK1) [36]. GPER activation also demonstrates anti-inflammatory effects by downregulation of interleukin (IL)-6 expression in macrophages by repression of NF-κB activity, through modulation of inhibitor of κB kinases (IKK) and c-jun N-terminal kinase (JNK) phosphorylation [37].

In summary, canonical activation of ERα promotes gene transcription of anti-oxidant enzymes, such as superoxide dismutase 2 (SOD2), and downregulates inflammatory proteins, like tumor necrosis factor (TNF)-α. Similarly, ERβ induces upregulation of proteins involved in vascular tone control (eNOS and cyclooxygenase-2) and downregulates proinflammatory and profibrotic gene programs. Membrane ERα and ERβ enhances NO production through eNOS phosphorylation and RhoA-associated kinase (ROCK) inhibition, and membrane ERβ also increases protein S-nitrosylation (SNO) via eNOS. Furthermore, GPER also increases NO production by eNOS phosphorylation and increases ERK/glycogen synthase kinase (GSK)-3β pathway activation while reduces TNF-α production and cyclin B1 and CDK1 activity (Figure 1). Therefore, the activation of ER in the cardiovascular system could be expected to reduce cardiac remodeling and the progression of HF induced by MI.

## 4. ER Modulation and MI: Preclinical Studies

Activation of cardiac ER leads to signaling pathways that culminate in cardioprotection, as summarized in Table 1 and Table 2. This protective role is generally thought to be based on an antioxidant effect and its consequent anti-atherogenic action [38]. However, the complexity of the cardiovascular actions of E2 has been extensively reported over the years. Female mice show a more expressive decrease in the rate of cardiac rupture during the acute phase of MI [39] and lower infarction area [40] than males. However, E2 does not improve ventricular dilatation or cardiac dysfunction in infarcted oophorectomized (OVX-MI) rats, and it even could increase mortality rate after MI [40], which led to an association of E2 to adverse effects and worst outcomes on post-MI cardiac remodeling.

Whereas short-term administration of E2 after MI would seem harmful, as it increases infarction area in OVX-MI rats, long-term treatment with E2 normalizes ventricular wall tension and chamber dimension in these animals by regulating the endothelin system and promoting increased survival [41]. Recent work has demonstrated that long-term E2 administration to OVX-MI rats failed to prevent cardiac dysfunction and remodeling [67]. Interestingly, OVX rats subjected to ischemia-reperfusion (I/R) present significantly decreased post-ischemic recovery of left ventricular function and increased MI extension, and these deleterious effects were avoided by E2 [68]. These divergences reinforce the idea that therapy onset and duration are determining factors for estrogen therapy outcomes.

In OVX-MI mice, activation of ERα reduces mortality in spite of increasing infarction extension, while stimulation of ERβ accounts for reduced infarction area and increased cardiac hypertrophy and mortality [69]. These effects are blocked by fulvestrant (ICI-182,780), an antagonist of both ERα and ERβ [70], and GPER agonist [9,71]. These findings will also prove useful to infarcted patients as E2 also reduces cardiomyocyte apoptosis in the infarction zone in OVX-MI mice [29] and the incidence and duration of arrhythmias observed in male dogs subjected to I/R [42]. These actions may rely on the inhibition of NF-κB-dependent gene expression [44], resulting in the reduction of myocyte death [45], as demonstrated by the anti-inflammatory and cardioprotective effects of the pathway-selective ER ligand, WAY-169916, after I/R in OVX rabbits [55].

Oral E2 therapy also has potential anti-atherogenic action besides anti-dyslipidemic and endothelium-dependent vasodilatory activity in postmenopausal women [72]. E2 exerts protective effects through activation of bone marrow-derived endothelial progenitor cells (EPCs), which infiltrate the infarcted myocardium and promote angiogenesis in OVX-MI mice [48,73]. These cells show increased migration, tubule formation, adhesion, and gene transcription activity in response to E2, which are mostly mediated by ERα, as validated by experiments with cells from knockout mice [48]. Moreover, EPCs display a greater abundance of *ESR1* gene transcripts than *ESR2* [48], and the increase in their ERα content and migratory capacity after incubation with E2 is blocked by the selective ERα antagonist, methylpiperidinopyrazole (MPP) [50]. Furthermore, VEGF is significantly downregulated in ERα knockout compared to wild-type EPC. However, the number of EPC and capillary density at the edge of infarcted zone are similar in both ERα and ERβ knockouts, indicating that both receptors contribute to E2-mediated EPC activation, tissue infiltration and preservation of cardiac function after MI, although ERα has a more prominent function. The SDF-1/CXCR4 pathway could participate in the effects of E2 in EPC, as the CXCR4 antagonist, AMD3100, reduces E2-induced EPC migration and this effect is further reduced in the presence of MPP [50].

Although some studies show the importance of estrogen and its receptors in post-MI angiogenesis, it has also been indicated that the absence of ERα and ERβ does not influence myocardial angiogenesis nor arteriogenesis after MI in mice from both sexes [74]. Moreover, E2 displays inhibitory effects on vascular calcification through nuclear ERα and inhibits inorganic phosphate-induced calcification and apoptosis of human VSMC by restoring Akt phosphorylation and Gas6 expression [75]. It is also important to highlight that E2 preserves vasodilator-response in epicardial coronary arteries of individuals with CAD, and small arterioles of these patients show similar reactivity to ERα and GPER [10], an additional beneficial effect for post-MI treatment.

After acute MI in male rats, there is a positive regulation of ERα but no change in ERβ expression in c-kit^+^ cardiac cells that accumulate in the peri-infarcted zone [76,77]. Post-MI isolated c-kit^+^ cells produce transcription factors involved in cardiogenic differentiation (GATA-4 and Notch-2) and cell renewal (Tbx3, Akt) [76,77]. When in co-culture with cardiomyocytes from MI rats, incubation of c-kit^+^ cardiac cells with E2 or propylpyrazoletriol (PPT), a selective ERα receptor agonist, increases myocyte survival, while the selective ERβ agonist, 2,3-bis-(4-hydroxyphenyl)-propionitrile (DPN) is devoid of action [77]. Additionally, membrane ERα also shows cardioprotective actions in long-term treatment with E2 in OVX-MI rats, such as attenuation of cardiac fibrosis by reducing RhoA/ROCK activation and cofilin phosphorylation [51].

The anti-apoptotic action of E2 seems to involve upregulation of ERα and modulation of insulin-like growth factor (IGF)-1 and 2, as the association of exenatide to low dose E2 prevents electrocardiographic and histological abnormalities, reduced serum cardiac markers and IGF-2 level in isoproterenol-induced MI in male rats [78]. However, higher E2 doses exerted a negative inotropic effect through attenuation of atrial essential myosin light chain (hALC-1) expression in AC16 cardiomyocytes [79]. The natural compound anthocyanin also reduces the risk of CVD by attenuating doxorubicin-induced cardiac apoptosis in male rats and H9c2 cells via ER by stabilizing heat shock factor 1 expression and decreasing IGF-2 receptor signaling [80]. In a MI model in male mice, activation of the ERα/PI3K/Akt/Notch1 pathway may also protect cardiomyocytes from oxidative damage and apoptosis, leading to reduced infarction area and serum cardiac markers and improved cardiac function [81].

There is evidence that the protective effect of estrogen on I/R could be related to its antioxidant action and nitric oxide (NO) production [82]. Endothelial expression of ERβ improves NO release during reperfusion and attenuates MI and vascular damage in male mice. Conditional overexpression of ERβ in the rat vascular wall also leads to increased expression of target genes, which modulate reactive oxygen species (ROS) generation, DNA damage, and mitochondrial function in both endothelium and cardiomyocytes [60]. Moreover, activation of both ERα and ERβ in human endothelial cells and high-fat diet-fed female mice up-regulates the expression of estrogen-related receptor α (ERRα), which is involved in fatty acid metabolism and mitochondrial function [83]. Activation of both ER also regulates mitochondrial SOD2 expression in endothelial cells, with ERβ controlling basal gene expression and ERα mediating gene transcription under stress conditions, thus reducing ROS production [47].

A direct cardiac action of ERβ has also been identified, as ERβ-deficient female mice exhibit increased I/R damage than wild-type female mice [52,62]. Activation of ERβ/eNOS axis promotes increased cardiac protein SNO in OVX mice treated with DPN or E2 and leads to a significantly better post-ischemic functional recovery and to reduced infarction area [53]. The SNO modification of cardiac proteins alter enzyme activity due to structural or functional changes in the protein and also protect against oxidation, contributing to cardiac protection in an I/R model in mice [84]. Membrane ER also induces the expression of myocyte-enriched calcineurin-interacting protein-1 (MCIP-1) via PI3K, which inhibits calcineurin [46] and reduces the expression of hypertrophy and remodeling genes in rat cardiomyocytes.

Although angiotensin II (AngII)-activated CF show similar expression and distribution of ER [85], ERβ plays an important role in reducing cell activation and cardiac fibrosis. Stimulation of ERβ in CF inhibits ROCK activity and transforming growth factor (TGF)-β production via AMPK and PKA activation [54], while it blocks c-Jun activity [27] and inhibits more than 90% of AngII-induced cardiac fibrosis in mice [63]. This effect is of great importance as AngII levels are elevated after MI in animal models [86] and patients [87], even those receiving angiotensin-converting enzyme inhibitors. In this latter case, the contribution of chymase-dependent AngII formation is suspected, as increased enzyme activity is observed in circulating leukocytes and myocardium after MI and seems to correlate with the severity of infarction [87]. In addition, worsening of diastolic function in OVX normotensive rats showed the greatest sensitivity to cardiac chymase activity [88], demonstrating the importance of cardiac ER activation in regulating AngII levels for additional improvement of cardiac function.

In addition, ERβ mediates cardioprotection in I/R on OVX mice through the PI3K/Akt pathway and anti-apoptotic signaling (decreased caspase-3 and -8 activities and increased Bcl-2) [64] and diminishes post-MI fibrosis [26,61]. ERβ could also influence arrhythmogenesis after MI, as infarcted females with ERβ deficiency show prolongation of ventricular repolarization and reduction in automaticity [65]. Thus, ERβ activation could reduce the incidence of ventricular tachyarrhythmia after MI. However, acute treatment with DPN had no effect on functional recovery after I/R in rats, regardless of gender or age [89]. These results would suggest either that ERβ signaling varies between different species or due to differences in the experimental protocol used. In addition, overexpression of ERβ led to improved survival, attenuated ventricular dilatation and hypertrophy, and improved cardiac function after MI in mice of both sexes, probably as a consequence of higher content of sarcoplasmic/endoplasmic reticulum Ca^2+^-ATPase 2a and reduced expression of collagens I and III, periostin and miR-21 [90].

Independently of cardiac ischemia or exercise, E2 exhibits beneficial protective effects against OVX-induced cardiac apoptosis by increasing ERβ-regulated survival pathways, such as IGF-1 and its receptor, Bcl-2 and phosphorylated PI3K and Bad, and decreasing expression of cell death-related proteins Bax, truncated Bid, cytochrome c, TNF-α and caspases-9 and -3 [91]. In addition, after I/R in OVX rats, PPT reduced plasma creatine kinase independently of exercise status, while in non-sedentary rats, both DPN and PTT reduced plasminogen activator inhibitor-1, TNF-α and interleukin (IL)-6 concentrations, indicating that a protective effect of both ERα and ERβ in I/R [92].

The role of GPER was investigated in atherosclerosis, a chronic vascular inflammatory process, and a risk factor for MI and peripheral stroke [23]. When submitted to an atherogenic diet, intact female mice with GPER deletion showed increased progression of atherosclerosis, higher levels of total cholesterol, low-density lipoprotein-cholesterol (LDL-C) and inflammatory markers, as well as a reduction in vascular response to NO. Estrogen depletion further accelerates the development of atherosclerosis, and treatment with G-1, a selective GPER agonist with similar affinity as E2, reduces atherosclerosis and inflammation with no uterotrophic effects [23]. Activation of GPER also contributes to LDL-C reduction by positively regulating LDL receptor expression in OVX-MI rats, possibly by downregulating pro-protein convertase subtilisin kexin type 9 (PCSK9) [93].

Under proinflammatory conditions, estrogen inhibits the production of vasoconstrictive prostanoids in endothelial cells through GPER activation [66], as indicated by increased carotid artery reactivity in both OVX and intact female mice with GPER deletion, as well as in porcine hearts. Furthermore, there are indications that G-1 and fulvestrant induce coronary vasodilation in a similar way but independently of ERα and ERβ [30]. These results suggest that GPER could be a therapeutic target to inhibit atherosclerosis and promote coronary vasodilation, reducing the risk of MI.

The risk of hypercholesterolemia and MI in the presence of estrogen depletion could be related to Ca^2+^ supplementation and a concomitant reduction of cAMP [56]. In estrogen depletion, long-term Ca^2+^ supplementation leads to increased intracellular Ca^2+^ concentration and phosphodiesterase activity in hepatocytes and subsequently reduces CYP7A1 content, which could limit hepatic cholesterol clearance. E2 and G-1 both control hepatic intracellular Ca^2+^ by transient receptor potential canonical-1 (TRPC1), and consequent blockade of plasma cholesterol increase by Ca^2+^ supplementation [56].

In parallel with studies on the benefits of GPER activation for reducing the risk of postmenopausal MI, the protective role of cardiac GPER was identified in both females and males. OVX aged rats present increased circulating levels of IL-1, IL-6, and TNF-α, which are exacerbated during I/R by mitochondrion-driven programmed necrosis through receptor-interacting protein (RIP)-1 signaling [94]. The cardiac expression of the *GPER* gene is significantly altered during I/R stress, increasing 2.4 times during hypoxia, and an additional 10.3 times during reoxygenation [20]. GPER activation during I/R improves cardiac functional recovery, reduces infarction extension in mice from both sexes, presumably by activating PI3K [17,20,22,57], and inhibits mitochondrial permeability transition pore (mPTP) opening in male mice via ERK [22]. The assumption that GPER is involved in cardioprotection via ERK was confirmed in a study using three different knockouts for ER (*ESR1*^−/−^, *ESR2*^−/−^ e *GPER*^−/−^) in male mice. It showed that only *GPER*^−/−^ hearts did not show E2-induced cardioprotection after I/R in relation to cardiac function, infarction size, and mitochondrial Ca^2+^ overload, suggesting that activation of MEK/ERK pathway via GSK-3β would be responsible for the beneficial effects [21]. More recently, it was shown that cardioprotection by E2 on I/R is blocked in mice of both sexes by G-15, a selective GPER antagonist, assuring the role of GPER activation [24].

Similar to the cardioprotective effect promoted by GPER in I/R, its activation with G-1 promotes the reduction of infarction size, myocardial fibrosis, and consequently, ventricular remodeling in OVX-MI females [58,59]. However, the improvement in cardiac remodeling in OVX-MI seems to involve both GPER and ERα, as the phosphorylation of Akt and eNOS observed after treatment with E2, G-1, or PPT is not totally blocked by G-15, fulvestrant, or actinomycin D [58]. These results indicate mechanisms relying on both GPER and membrane ERα, which converge to activation of PI3K/Akt/eNOS. OVX-MI animals treated with G-1 showed improved cardiac function by decreased apoptosis and reduced myocardial inflammation, as indicated by lower TNF-α and higher IL-10 levels. The reduction in cardiomyocyte apoptosis could probably be related to the activation of the PI3K/Akt pathway, as already shown for serum-deprived hypoxic CF incubated with G-1 [59]. Complementing the beneficial cardiac action of this receptor, GPER activation also regulates myocardial Ca^2+^ dynamics, left ventricular pressure, and the frequency of ectopic contractions in isoproterenol-induced MI model in female rats and prevents potentiation of peak currents through L-type Ca^2+^ channels in cardiomyocytes [95].

In addition to ER agonists and antagonists, studies with selective ER modulators (SERM), substances with complex activities in the ER, such as tamoxifen and raloxifene, were also performed. Tamoxifen induces vasodilation in porcine coronary arteries with rapid onset, similarly to E2 and presumably by non-genomic action, explaining the beneficial impact of the use of tamoxifen in atherosclerotic vascular disease [96]. However, tamoxifen-treated female rats show increased damaged area after MI, in part by reduction of angiogenesis and inhibition of endothelial cell migration and proliferation by suppression of VEGF [97]. Pre-treatment of female rats with tamoxifen reduced levels of total cholesterol, LDL-C and very low-density cholesterol (VLDL-C), triglycerides, reduced heart attack, cardiac hypertrophy, and prevented oxidative stress in rats with isoproterenol-induced MI [98]. However, totally different behavior is observed when treatment is initiated post-MI, which indicates tamoxifen has better preventive than therapeutic effects.

Long-term raloxifene treatment does not activate ERα, but activates GPER instead [70,99], reducing MI-induced arrhythmias and attenuating cardiac damage and apoptosis after I/R in OVX rats, which correlates to lower neutrophil infiltration and suppression of NF-κB activity [100]. Raloxifene restores acetylcholine-induced vasodilation and attenuates vasoconstriction induced by thromboxane receptor agonist in arteries of OVX hamsters fed a cholesterol-rich diet [101]. Raloxifene also increases heme oxygenase activity and inhibits myeloperoxidase in OVX animals, promoting vasodilation and preventing elevation of the ST segment, the isoelectric period between depolarization and repolarization on the electrocardiogram, further indicating a cardioprotective role associated with antioxidant and anti-inflammatory actions [102]. Hence, SERMs can be useful for preserving endothelial function and vascular hypersensitivity in coronary arteries in patients with hypercholesterolemia or hyperlipidemia involved in MI.

Therefore, E2 displays a broad range of cardioprotective effects and can be useful for the treatment of CVD, especially MI. Besides anti-dyslipidemic, anti-atherogenic, and vasodilator effects, E2 may exert direct cardiac effects, such as increasing cell survival, stimulating ischemic preconditioning, regulating myocardial hypertrophy and fibrosis, as well as normalizing cardiac function and rhythm. Activation of ER plays an important role in most E2 effects, with an interplay of ER subtype-selective actions generally seen in both sexes. Among them, ERα stimulates myocardial regeneration and angiogenesis through activation of c-kit^+^ cardiac cells and EPC and improves antioxidant mechanisms and cardiac function while reduces VSMC calcification, coronary tonus, fibrosis, and post-MI mortality. In general, ERβ stimulation leads to upregulation of antioxidant and anti-apoptotic gene programs, although its role in attenuating fibrosis and arrhythmias has already been shown. Besides, GPER activation generates not only coronary vasodilation and controls cholesterol catabolism and inflammation but also improves cardiac conditioning after I/R injury. Moreover, ERα and GPER synergistically contribute to reduction in infarct size and ventricular remodeling, and these beneficial actions prompt further research on estrogen effects in MI patients, as subsequently discussed.

## 5. ER Modulation and MI: Clinical Studies

In 2017, heart disease accounted for 24.2% and 21.8% of all male and female deaths worldwide, respectively. Ischemic heart disease and HF are the main contributors to mortality, and gender difference stills an important issue as HF contributes more to CVD-related mortality in women than men, mainly due to undiagnosed ischemic heart disease [103].

Compared to men, women suffer from ischemic heart disease at an older age, and this could be explained by the protective effect of endogenous estrogens [104], as premature natural or surgical menopause is also associated with increased risk of CVD [105]. OVX aggravates long-term health issues such as cardiovascular disease (CVD) incidence and mortality, especially if estrogen depletion occurs before 45–50 years old [106,107]. Although previously existing CVD risk factors may influence menopause age, a recent study and meta-analysis demonstrated a higher incidence of HF in women who experienced menopause before the age of 45, even after adjusting for possible confounders [108].

Likewise, a prospective study demonstrated that women under 40 years old with premature ovarian insufficiency (POI) showed a 50–70% higher risk of death by ischemic heart disease when compared with those reaching menopause by 50 [109,110,111]. In addition, POI seems to predispose to electrocardiographic alterations [112] with reduced cardiac automaticity [108]. When compared to healthy women, patients not on E2 therapy have impaired diastolic function and myocardial performance index [113]. Moreover, endothelial dysfunction, increased carotid intima-media thickness, and disturbances in lipid profile observed in POI are all known risk factors for CVD [114]. In addition, inverse relationships were also verified between the age of menopause onset and the risk of HF or ischemic heart disease, and OVX before 45 years old is generally associated with higher risks [115].

Besides, a prospective study of women under 50 years old demonstrated a positive correlation between vasomotor symptoms and coronary intima-media thickness. Moreover, menopausal women over 50 with early-onset vasomotor symptoms also show increased CVD mortality risk when compared to women with later onset [110]. Furthermore, a marked increase in CVD-associated mortality is also seen in that first group if no E2 supplementation was taken after menopause onset [109].

Thus, as a result of such disparities driven by sex or estrogen status, many clinical studies were conducted in order to examine the effects of hormone replacement therapy (HRT) on the cardiovascular system. A significant reduction in myocardial hypertrophy was observed in postmenopausal women receiving HRT, suggesting a role for ER modulation in the development of hypertrophy and HF. However, according to Women’s Health Initiative (WHI), the association of estrogen and progesterone does not reduce the incidence of stroke, CAD, and pulmonary embolism in menopausal women over 50 years old, implying the risks outweigh the benefits for this therapeutic approach and treatment should not be started or continued for primary prevention of CAD [116]. In addition, an observational study suggested HRT provides little or no benefit to postmenopausal women in both primary and secondary prevention of cardiovascular events and also increases the risk of stroke and venous thromboembolism [117].

Contrary to these findings, a meta-analysis including 23 controlled and randomized clinical studies indicated that women under 60 years old or ten years of menopause onset experience a 32% reduction in CAD incidence by HRT, while women of older age or over ten years of menopause onset do not benefit from this treatment. [118]. Similarly, a five-year follow-up of postmenopausal women demonstrated that oral E2 was associated with reduced progression of subclinical atherosclerosis if therapy was started six but not ten or more years after menopause onset [106]. A recent review of clinical trial data also suggested HRT could be used as primary prevention for CVD, as women close to menopause undergoing HRT had decreased risk of CAD [107].

Manson et al. (2019) observed that adverse effects seen after treatment with estrogen were more significant in women over 70 years old, while OVX women aged 50–59 years gained a reduction in all-cause mortality [119]. In addition, among older postmenopausal women with CVD, estrogen therapy associated with progestogen can substantially increase the risk of CVD events in the first year of treatment among women with clinically significant flushing [120]. E2 levels were also differently associated with the progression of atherosclerosis according to the time of initiation of HRT. Higher levels of E2 decreased the rate of progression of carotid intima-media thickness in early postmenopausal women but increased it in women in late menopause. These facts also support the “Timing Hypothesis” of HRT on reducing atherosclerosis [121,122].

However, HRT or its initiation time showed no significant effect on atherosclerosis measurements in postmenopausal women, such as coronary artery calcium, total stenosis, and plaque in any of the postmenopausal stratum [106]. Furthermore, coronary artery calcification shows a strong correlation to incident CAD, CVD, and total mortality over about eight years in postmenopausal women aged 50–59 years, regardless of estrogen therapy or risk factors [123]. Although none of these studies examined the impact of HRT on cardiac function, its involvement is quite possible, as ischemia precedes congestive diastolic and systolic HF [124].

Despite conflicting results on CVD prevention, thrombotic complications still a major concern of cardiovascular risk from HRT, and yet oral formulations may predispose women to venous thromboembolism; this risk seems to be reduced with transdermal patches, as well as for deep vein thrombosis [107,125]. A systematic review by Bezwada et al. (2017) also highlighted some positive evidence regarding the cardioprotective effect of transdermal estrogen but pointed out the need for further randomized controlled studies with longer follow-up for drawing solid conclusions [126].

Vascular effects of E2 may also depend on the route of administration, as improvement of vascular function could be observed after oral treatment in 27 postmenopausal women but not when patients received transdermal E2 or placebo. Oral E2, but not transdermal, also induced significant decreases in LDL-C and lipoprotein A (LpA) concentrations, as well as increased high-density lipoprotein cholesterol (HDL-C) in two weeks [72]. As E2 reduces LDL-C oxidation in postmenopausal women, this antioxidant effect seems to contribute to its anti-atherogenic action [38]. However, oral administration of a high-dose of estrogen, associated or not with progestogen, may increase the risk of hypertension in aged postmenopausal women, and different estrogen formulations may offer lower risks [127].

Spontaneous menopause can induce metabolic syndrome and insulin resistance but is not associated with an increased risk of diabetes in advancing age. HRT may result in neutral or beneficial effects on glucose levels in patients with pre-existing type 2 diabetes, although oral treatment has superior performance than transdermal estrogen [128].

Although often underestimated, hot flushes and bone loss are also observed in prostate cancer patients undertaking androgen deprivation therapy, possibly caused by E2 deficiency [129,130]. Despite a lack of relationship with estrogen status, these patients showed a higher risk of nonfatal and fatal ischemic heart disease and MI [131]. Further analysis would be necessary to demonstrate if treatment of low dose E2 could be of benefit for reducing adverse effects of cancer therapy and mortality [132,133].

Among SERM, the use of tamoxifen by women is associated with a lower incidence of MI and vascular disease-associated mortality. As a partial agonist, tamoxifen exerts coronary vasodilation, improving blood flow in the myocardium, and is associated with reduced deaths, lower rates of MI, and hospitalization due to heart disease [134]. In the surveillance, epidemiology, and end results (SEER)-Medicare database population study, women over 67 years with breast cancer had no significant differences in MI risk when treated with tamoxifen and aromatase inhibitors [135]. However, the risk of cardiovascular events is lower in postmenopausal women with breast cancer after treatment with tamoxifen when compared to aromatase inhibitors [136,137].

Raloxifene also has favorable effects on total cholesterol and LDL-C levels, in addition to reducing fibrinogen and LpA and increasing HDL-C [138]. HRT and raloxifene have been found to reduce serum homocysteine levels [139] and improve vascular endothelial function in postmenopausal women [140]. In addition, raloxifene therapy for four years significantly reduced the risk of cardiovascular events in women at increased risk before treatment, and its use does not increase the risk of early cardiovascular events [141]. Toremifen, another second-generation SERM with similar affinities for ERs as tamoxifen and raloxifene [142], significantly improved serum lipid profile in postmenopausal women [82].

As ER modulators of plant origin, soy isoflavones, when administered to women for six months from menopause onset, also reduce metabolic parameters and systolic arterial pressure, lowering by 37% the risk of MI, 24% of CVD, 27% the ten-year risk of CAD, and 42% the risk of cardiovascular death [143,144].

Despite an association between estrogen depletion and dyslipidemia have already been established, Ca^2+^ supplementation seems to accelerate serum cholesterol level elevation and to increase the risk of hypercholesterolemia and MI in menopausal women, possibly due to decreased catabolism of cholesterol, as observed in an OVX rat model [56]. Interestingly, according to WHI, women submitted to vitamin D and Ca^2+^ supplementation one year after starting HRT showed a 50% reduction in the risk of stroke if undertaking estrogen-only therapy, while its association to progesterone showed no effect [145].

Thus, as evidenced by most recent data, the benefits of HRT in reducing the risk of CVD in postmenopausal women seem to be related to the moment of therapy onset, being evident in the initial phase of menopause. In addition, the route of administration may also influence the outcomes and, while transdermal E2 patches are less likely to induce thrombotic events, oral E2 formulation provides an improvement in serum lipid profile and risk of atherosclerosis. It was also shown that SERM reduces the rate of MI and hospitalizations for heart disease and may also improve lipid profile in women at increased risk of CVD. In this way, women undertaking HRT or using SERM could benefit from an improved quality of life and lower risks of CVD.

## 6. Genetic Factors Related to Estrogen Receptors and MI

Estrogen and related HRT activate ER, which in turn regulate gene transcription of proteins associated with CVD risk factors. Individuals with variations in *ESR1* are more susceptible to cardiovascular complications, including ischemic heart disease and MI. In a prospective study including 1739 participants, 49.7% women, two *ESR1* polymorphisms were detected (*ESR1* c.454-397>C and *ESR1* c.454-351>G), and it was suggested that recessive individuals for the C allele (nearly 20% of the cohort population) were more likely (at a two-fold rate) to develop CVD including MI [146]. More recently, it was reported that men bearing the genotype *ESR1* c.454-397CC could also be at risk for stroke [147]. In a population-based prospective cohort study involving 6408 participants, 59.2% women, polymorphisms c.454-397T>C and c.454-351A>G were observed in the *ESR1*, with the possibility of four haplotype alleles and during a seven-year follow-up, 285 patients had MI and 440 had ischemic heart disease. These studies found that only postmenopausal women who carry *ESR1* haplotype 1 (c.454-397T and c.454-351A alleles) had an increased risk of MI and ischemic heart disease, regardless of known cardiovascular risk factors, while no correlation was observed for men [148]. Women with a variant phenotype CC of *ESR1* (c.454-397T>C) also had thicker coronary intima compared to women with TT genotype, pointing to the importance of *ESR1* genotype in susceptibility to CVD [149]. The -351A/G *ESR1* polymorphism may be associated with MI, but not with extreme longevity, as the AA genotype (-351A/G) is more frequent in women with MI than in aged women with no history of MI or young patients. Moreover, in MI patients, the GG genotype (-351A/G) was associated with more favorable levels of total cholesterol, LDL-C, and HDL-C compared to the AG and AA genotype [150].

Considering the *ESR2* gene, single nucleotide polymorphisms could affect the susceptibility to MI in a sex-dependent manner. The rs1271572T allele was significantly more common in patients who developed MI being more pronounced in men, as well the T variant was associated with an increased risk of MI in the Spanish population and was considered limited to men only [151]. Another variant of the *ESR2* gene, G1730A, was also related to a higher incidence of MI in younger men, while no correlation was found between this variant and the occurrence of MI in women [152]. ESR2 polymorphisms may also be associated with an increased risk of metabolic syndrome in postmenopausal women [153] and contribute to salt-sensitive blood pressure in premenopausal women [154].

Given their most recent discovery relative to ERα and ERβ, CVD-related *GPER* polymorphisms have not been thoroughly investigated although associations were already demonstrated to idiopathic scoliosis [155], the risk for seminoma [156] and uterine leiomyoma [157] and more recently, with risk and development of hypertension [158]. Moreover, a hypofunctional *GPER* variant P16L has recently been identified and associated with increased LDL-C in plasma, which may pose an additional risk of atherosclerosis [93].

ER gene variations are important factors associated with a risk for CVD in a sex-dependent manner and may possibly determine pharmacological responses for ER-targeted therapies. While ERS1 variations in women show correlations with increased risk for CVD, including MI, ERS2 polymorphisms may increase MI susceptibility in men. However, ER gene variants may also predispose to increased risk factors for CVD, as seen in women bearing ERS2 variants or a hypo-functional GPER variant.

## 7. Conclusions

Despite the extensive knowledge on the cardioprotective effects of estrogen, clinical studies conducted on MI and CVD still include mixed, favorable, and unfavorable profiles. One of the possible shortcomings of those is the use of drugs for HRT treatment with low selectivity for the three ER subtypes, unlike pharmacological agents used in preclinical research. In addition, the experimental design would also have influenced the clinical results as, in general, most of the evaluated women started treatment years after menopause onset, and very few included male subjects, while preclinical studies often assess both males and females with or without estrogen depletion. The existence of genetic variants and gene polymorphisms on ER genes is another extremely relevant factor, which should inspire caution when attempting to translate the beneficial effects of the activation of different ERs for the treatment of cardiac diseases in rodents and humans directly.

In fact, it is undeniable that ER modulation causes a wide range of effects on the cardiovascular system, the activation of “classic” ER and GPER demonstrate beneficial properties that could be explored for the treatment of MI and consequent prevention of HF. Further research should address questions such as the elucidation of molecular mechanisms underlying cardiovascular estrogen actions, as well as the influence of clinical features like age, menopause onset, genetic background, and presence of risk factors on the beneficial cardiovascular effects observed during estrogen therapy and ER activation.

## Figures and Tables

**Figure 1 ijms-22-00525-f001:**
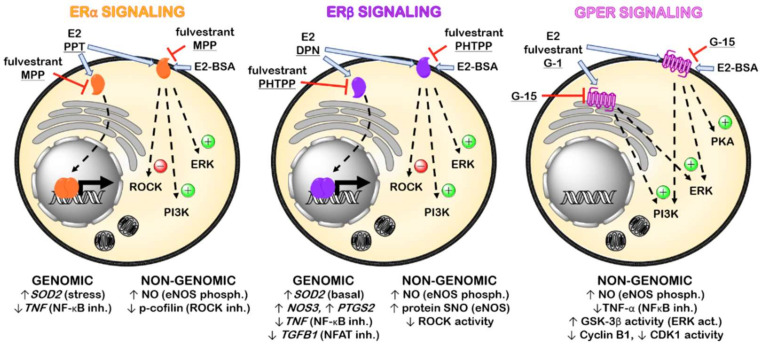
Representative scheme of the main pharmacological agents targeting ER and signaling pathways involved in the mechanism of action of their genomic and non-genomic effects. (Act., activation; Inh., inhibition; Phosph., phosphorylation. Underlined: ER subtype-selective agonists and antagonists).

**Table 1 ijms-22-00525-t001:** Investigation of estrogen receptor (ER) actions in myocardial infarction (MI) using pharmacological approaches.

Agonist	Receptor	Model	Effect	Mechanism	Ref.
E2	Unknown	acute and chronic OVX-MI—Rats	↑ infarcted area ↓ ventricular wall tension ↓ ventricular dilatation ↑ survival	↑ endothelin B receptor	[41]
E2	Unknown	I/R—Male dogs	↓ infarcted area ↓ arrhythmias	opens K^+^ channels ↑ NO production	[42]
E2	Unknown	OVX-MI—Mice	↓ cardiomyocyte apoptosis ↓ infarcted area	PI3K/Akt activation	[29,43]
E2	Unknown	Staurosporine-induced apoptosis—Rat cardiomyocytes	↓ cardiomyocyte apoptosis	NF-κB inhibition	[44,45]
E2	Unknown	AngII-induced hypertrophy—Rat cardiomyocytes	↓ hypertrophy and remodeling gene transcription	calcineurin inhibition (via PI3K)	[46]
E2	ERβ (ERα if under stress)	OVX + high-fat diet—Mice	Improves cardiovascular function	↑ SOD2 (heart and aorta)	[47]
E2	ERα > ERβ	acute and chronic OVX-MI—Mice	↓ injured myocardium ↑ angiogenesis	activation of bone marrow-derived EPC	[48,49,50]
E2, E2-BSA, PPT	ERα	OVX-MI—Rats	↓ cardiac fibrosis	↓ cofilin phosphorylation (↓ RhoA/ROCK activity)	[51]
E2, DPN	ERβ	OVX + I/R—Mice	↓ I/R myocardial injury	↑ cardioprotective genes (COX2, GADD45β, IL-6, heat shock proteins, PFKFB)	[52]
E2, DPN	ERβ	OVX + I/R—Mice	Improves cardiac function ↓ infarcted area	↑ cardiac protein SNO	[53]
G1	GPER	OVX + atherogenic—Mice	↓ atherosclerosis ↓ vascular inflammation	↓ vessel inflammation ↑ NO production	[23]
E2, βLGND, DPN	ERβ	AngII-induced fibrosis—Rat cardiac fibroblasts	↓ extracellular matrix production	↓ ROCK activity ↓ TGF-β production (via AMPK and PKA)	[54]
E2, WAY-169916	ERα/ERβ	OVX + I/R—Rabbit	↓ infarcted area ↓ neutrophil accumulation	NF-κB inhibition	[55]
E2, G-1	GPER	OVX + Ca^2+^ supplementation—Rats	↓ total cholesterol	hepatic TRPC1 inhibition, ↓ intracellular Ca^2+^	[56]
G-1	GPER	I/R—Male mice, female and male rats	Improves cardiac function ↓ infarction area	↓ oxidative stress ↓ ATP depletion ↓ mPTP opening (via PI3K/Akt)	[20,22,57]
E2, PPT, G-1	ERα/GPER	OVX-MI—Rats	↓ infarction size ↓ myocardial fibrosis	↑ eNOS activity (via PI3K/Akt)	[58]
G-1	GPER	OVX-MI—Mice	Improved cardiac function ↓ apoptosis ↓ myocardial inflammation	↓ TNF-α ↑ IL-10 (via PI3K/Akt)	[59]

βLGN2, ERβ agonist; BSA, bovine serum; COX2, cyclooxygenase 2; eNOS, endothelial nitric oxide synthase; EPC, endothelial progenitor cells; DPN, 2,3-bis-(4-hydroxyphenyl)-propionitrile; E2, 17β-estradiol; ER, estrogen receptor; GPER, G protein-coupled estrogen receptor; GADD45β, growth arrest and DNA damage 45β; IL-10, interleukin 10; IL-6, interleukin 6; I/R, ischemia-reperfusion; mPTP, mitochondrial permeability transition pore; MI, myocardial infarction; NO, nitric oxide; NF-κB, nuclear factor kappa B; OVX, oophorectomy; PFKB, 6-phosphofructo-2kinase/fructose-2,6-bisphosphatase; PI3K, phosphatidylinositol 3-kinase; PPT, propylpyrazoletriol; SNO, *S*-nitrosylation; SOD, superoxide dismutase; TGF-β, transforming growth factor-β; TRPC1, transient receptor potential canonical-1; TNF-α, tumor necrosis factor-α.

**Table 2 ijms-22-00525-t002:** Investigation of ER actions in MI using molecular biology approaches.

Strain	Model	Phenotype	Mechanism	Ref.
Targeted endothelial ErβOE—Male rats	I/R	↓ oxidative burst ↓ vascular injury	↑ ERRα, SOD2, eNOS expression ↑ NO/ROS ratio ↑ mitochondrial function	[60]
Targeted cardiac ErβOE—Female mice	MI	↓ ventricle dilatation ↓ fibrosis	↓ fibrotic gene program expression	[61]
ERβKO—Female and male mice	I/R	↑ cardiac injury after isoproterenol (females)	↓ recovery of intracellular ATP level ↑ intracellular acidification ↓ eNOS, lipogenic enzymes	[62]
ERβKO—Mice	OVX + AngII	↓ E2 anti-hypetrophy and anti-fibrosis effects	↓ PI3K/MCIP1 inhibition of hypertrophy and fibrosis gene programs	[26,63]
ErβKO—Mice	OVX + I/R	↑ cardiac damage	↓ PI3K/Akt pathway ↑ apoptotic signaling	[64]
ErβKO—Female mice	MI	↑ ventricular repolarization time ↓ ventricular premature beats	↓ mRNA levels of K_v_4.3	[65]
GPERKO—Mice	OVX and atherogenic diet	↑ atherosclerosis progression	↑ vascular inflammation ↓ vascular basal NO production	[23]
GPERKO—Mice	OVX and atherogenic diet	↓ anti-atherogenic action of E2 ↓ coronary vasodilatation	↑ endothelium-dependent contractions to vasoconstrictor prostanoids	[66]
GPERKO—Male mice	I/R	↓ E2-improvement in cardiac function ↑ infarcted area ↑mitochondrial Ca^2+^ overload	↓ activation of MEK/ERK pathway GSK-3β	[21]

α-SMA, α-smooth muscle actin; E2, 17β-estradiol; eNOS, endothelial nitric oxide synthase; ER, estrogen receptor; ERRα, estrogen-related receptor alpha; ERαKO ERα knockout; ERβKO, ERβ knockout; ERβOE, ERβ overexpression; GPER, G protein-coupled estrogen receptor; GPERKO, GPER knockout; GSK-3β, glycogen synthase kinase 3β; I/R, ischemia-reperfusion; MCIP, myocyte-enriched calcineurin-interacting protein-1; MI, myocardial infarction; NFAT, nuclear factor of activated T cells; OVX, oophorectomy; PI3K, phosphatidylinositol 3-kinase; ROS, reactive oxygen species; SOD, superoxide dismutase; TGF-β, transforming growth factor; WT, wild-type.

## Data Availability

Not applicable.

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
