# Peer review of "Estrogen Receptors: Therapeutic Perspectives for the Treatment of Cardiac Dysfunction after Myocardial Infarction"

_ijms, 2021, doi:10.3390/ijms22020525_

Round 1

Reviewer 1 Report

It is an interesting topic and generally interesting review. I have some suggestions for consideration:

  1. In the introduction, the authors seem to emphasize the importance of heart failure, whilst the following content did not include much content about heart failure. It is quite confusing to me.
  2. The review included quite a lot about the mechanisms and pre-clinical work, but limited clinical studies. Actually there are quite a lot of epidemiological work on estrogen and CVD, which should be included in the review.
  3. The opinion about different timing of estrogen and CVD contributes to differences in different studies seems to be one-sided. There are other explanations, such as estrogen supplementation is relevant to socioeconomic position, and the association between estrogen and CVD might be confounded. These should also be included in the review.
  4. I think the figure about genomic and nongenomic is quite confusing and not linked to the relevant content.

Author Response

Reviewer Report 1

Comments and Suggestions for Authors

It is an interesting topic and generally interesting review. I have some suggestions for consideration:

Comment 1: In the introduction, the authors seem to emphasize the importance of heart failure, whilst the following content did not include much content about heart failure. It is quite confusing to me.

Response: Since the focus of this review is not heart failure (HF) itself, but the prevention of its development, as result of myocardial infarction, it is clear the lack of discussion about HF during the review. However, for a better understanding of this proposal, it was included the following paragraph:

Line 39:

“Myocardial infarction (MI) remains the most common cause of heart failure (HF), despite notable advances in the treatment of coronary artery disease (CAD) and MI over the past two decades (7). Thus, early treatment of HF induced by MI is an important clinical strategy for a better prognosis. Although HF is the leading cause of death regardless of sex, women tend to develop HF later and have a better prognosis.”

7) Jenča, D., Melenovský, V., Stehlik, J., Staněk, V., Kettner, J., Kautzner, J. Wohlfahrt, P. (2020). Heart failure after myocardial infarction: incidence and predictors. ESC Heart Failure, ehf2.13144. https://doi.org/10.1002/ehf2.13144

Comment 2: The review included quite a lot about the mechanisms and pre-clinical work, but limited clinical studies. Actually, there are quite a lot of epidemiological work on estrogen and CVD, which should be included in the review.

Response: As suggested by the reviewer, it was included the following paragraphs:

Line 368:

“OVX aggravates long-term health issues such as cardiovascular disease (CVD) incidence and mortality, especially if estrogen depletion occurs before 45-50 years old (106,107). Although previously existing CVD risk factors may influence menopause age, a recent meta-analysis study demonstrated a higher incidence of heart failure (HF) in women who experienced menopause before the age of 45, even after adjusting for possible confounders (108). Likewise, a prospective study demonstrated that women under 40 years old with premature ovarian insufficiency (POI) show 50-70% higher risk of death by ischemic heart disease, when compared with those reaching menopause after 50 (109, 110, 111). In addition, POI seems to predispose to electrocardiographic alterations (112) with reduction of cardiac automaticity (108). When compared to healthy women, patients not on E2 therapy have impaired diastolic function and myocardial performance index (113). Moreover, endothelial dysfunction, increased carotid intima-media thickness and disturbances in lipid profile observed in POI, are all known risk factors for CVD (114). In addition, inverse relationships were also verified between age of menopause onset and the risk of HF or ischemic heart disease and OVX before 45 years old is generally associated with higher risks (115). Besides, a prospective study of women under 50 years old demonstrated a positive correlation between vasomotor symptoms and coronary intima-media thickness. Moreover, menopausal women over 50 with early onset vasomotor symptoms also show increased CVD mortality risk when compared to women with later onset (110). Furthermore, a marked increase in CVD-associated mortality is also seen in that group who never used E2 after menopause onset (109).”

106) Hodis, H. N., Mack, W. J., Henderson, V. W., Shoupe, D., Budoff, M. J., Hwang-Levine, J., … ELITE Research Group. (2016). Vascular Effects of Early versus Late Postmenopausal Treatment with Estradiol. The New England Journal of Medicine, 374(13), 1221–1231. https://doi.org/10.1056/NEJMoa1505241

107) Marko, K. I., & Simon, J. A. (2018). Clinical trials in menopause. Menopause (New York, N.Y.), 25(2), 217–230. https://doi.org/10.1097/GME.0000000000000978

108) Appiah, D.; Schreiner, P.J.; Demerath, E.W.; Loehr, L.R.; Chang, P.P.; Folsom, A.R. Association of Age at Menopause With Incident Heart Failure: A Prospective Cohort Study and Meta‐Analysis. J. Am. Heart Assoc. 2016, 5, doi:10.1161/JAHA.116.003769.

109) Jacobsen, B.K.; Knutsen, S.F.; Fraser, G.E. Age at Natural Menopause and Total Mortality and Mortality from Ischemic Heart Disease. J. Clin. Epidemiol. 1999, 52, 303–307, doi:10.1016/S0895-4356(98)00170-X.

110) Hamoda, H. (2017). The British Menopause Society and Women’s Health Concern recommendations on the management of women with premature ovarian insufficiency. Post Reproductive Health, 23(1), 22–35. https://doi.org/10.1177/2053369117699358

111) Okoth, K.; Chandan, J.S.; Marshall, T.; Thangaratinam, S.; Thomas, G.N.; Nirantharakumar, K.; Adderley, N.J. Association between the reproductive health of young women and cardiovascular disease in later life: umbrella review. BMJ 2020, m3502, doi:10.1136/bmj.m3502.

112) Canpolat, U.; Tokgözoğlu, L.; Yorgun, H.; Bariş Kaya, E.; Murat Gürses, K.; Şahiner, L.; Bozdağ, G.; Kabakçi, G.; Oto, A.; Aytemir, K. The association of premature ovarian failure with ventricular repolarization  dynamics evaluated by QT dynamicity. Eur.  Eur. pacing, arrhythmias, Card. Electrophysiol.  J.  Work. groups Card. pacing, arrhythmias, Card. Cell. Electrophysiol. Eur. Soc. Cardiol. 2013, 15, 1657–1663, doi:10.1093/europace/eut093.

113) Aksoy, M.N.M.; Akdemir, N.; Kilic, H.; Yilmaz, S.; Akdemir, R.; Gunduz, H. Pulse wave velocity and myocardial performance index in premature ovarian  insufficiency. Scand. Cardiovasc. J. 2017, 51, 95–98, doi:10.1080/14017431.2017.1285044.

114) Podfigurna-Stopa, A.; Czyzyk, A.; Grymowicz, M.; Smolarczyk, R.; Katulski, K.; Czajkowski, K.; Meczekalski, B. Premature ovarian insufficiency: the context of long-term effects. J. Endocrinol. Invest. 2016, 39, 983–990, doi:10.1007/s40618-016-0467-z.

115) Paciuc, J. Hormone Therapy in Menopause. In Hormonal Pathology of the Uterus. Advances in Experimental Medicine and Biology, vol 1242; Deligdisch-Schor, L., Miceli, A.M., Eds.; Springer, Cham, 2020; pp. 89–120.

Line 442:

Although often underestimated, hot flushes and bone loss are also observed in prostate cancer patients undertaking androgen deprivation therapy, possibly caused by estradiol (E2) deficiency (130,131). Despite a lack of relationship with estrogen status, these patients show higher risk of nonfatal and fatal ischemic heart disease and MI (132). Further analysis would be necessary to demonstrate if treatment of low dose E2 could be of benefit for reducing cancer therapy adverse effects and mortality (133,134).

130) Frisk, J. Managing hot flushes in men after prostate cancer—A systematic review. Maturitas 2010, 65, 15–22, doi:10.1016/j.maturitas.2009.10.017.

131) Freedland, S.J.; Eastham, J.; Shore, N. Androgen deprivation therapy and estrogen deficiency induced adverse effects in the treatment of prostate cancer. Prostate Cancer Prostatic Dis. 2009, 12, 333–338, doi:10.1038/pcan.2009.35.

132) Bosco, C.; Bosnyak, Z.; Malmberg, A.; Adolfsson, J.; Keating, N.L.; Van Hemelrijck, M. Quantifying Observational Evidence for Risk of Fatal and Nonfatal Cardiovascular Disease Following Androgen Deprivation Therapy for Prostate Cancer: A Meta-analysis. Eur. Urol. 2015, 68, 386–396, doi:10.1016/j.eururo.2014.11.039.

133) Russell, N.; Hoermann, R.; Cheung, A.S.; Ching, M.; Zajac, J.D.; Handelsman, D.J.; Grossmann, M. Short-term effects of transdermal estradiol in men undergoing androgen deprivation therapy for prostate cancer: a randomized placebo-controlled trial. Eur. J. Endocrinol. 2018, 178, 565–576, doi:10.1530/EJE-17-1072.

134) Gilbert, D.C.; Duong, T.; Kynaston, H.G.; Alhasso, A.A.; Cafferty, F.H.; Rosen, S.D.; Kanaga-Sundaram, S.; Dixit, S.; Laniado, M.; Madaan, S.; et al. Quality-of-life outcomes from the Prostate Adenocarcinoma: TransCutaneous Hormones (PATCH) trial evaluating luteinising hormone-releasing hormone agonists versus transdermal oestradiol for androgen suppression in advanced prostate cancer. BJU Int. 2017, 119, 667–675, doi:10.1111/bju.13687.

Comment 3: The opinion about different timing of estrogen and CVD contributes to differences in different studies seems to be one-sided. There are other explanations, such as estrogen supplementation is relevant to socioeconomic position, and the association between estrogen and CVD might be confounded. These should also be included in the review.

Response: As suggested by the reviewer the following paragraphs were included in the text:

Line 361:

In 2017, heart disease accounted for 24.2% and 21.8% of all male and female deaths worldwide, respectively. Ischemic heart disease and HF are the main contributors to mortality and the gender difference is important in this respect because HF is responsible for mortality in women, mainly due to an undiagnosed ischemic heart disease (103). Compared to men, women suffer from ischemic heart disease at an older age, which could be explained by the protective effect of endogenous estrogens (104), as it is already shown that premature natural and surgical menopause is also associated with a significantly increased risk of CVD (105).

103) Mauvais-Jarvis, F., Bairey Merz, N., Barnes, P. J., Brinton, R. D., Carrero, J.-J., DeMeo, D. L., … Suzuki, A. (2020). Sex and gender: modifiers of health, disease, and medicine. The Lancet, 396(10250), 565–582. https://doi.org/10.1016/S0140-6736(20)31561-0

104) Mehta, L. S., Beckie, T. M., DeVon, H. A., Grines, C. L., Krumholz, H. M., Johnson, M. N., … Wenger, N. K. (2016). Acute Myocardial Infarction in Women. Circulation, 133(9), 916–947. https://doi.org/10.1161/CIR.0000000000000351

105) Honigberg, M. C., Zekavat, S. M., Aragam, K., Finneran, P., Klarin, D., Bhatt, D. L., Natarajan, P. (2019). Association of Premature Natural and Surgical Menopause With Incident Cardiovascular Disease. JAMA, 322(24), 2411. https://doi.org/10.1001/jama.2019.19191

Line 407:

“Manson et al. (2019) observed that adverse effects seen after treatment with estrogen were more significant in women over 70 years old, while OVX women aged 50-59 years benefited of a reduction in all-cause mortality (120). In addition, among older postmenopausal women with CVD, estrogen therapy associated to progestogen can substantially increase the risk of CVD events in the first year of treatment among women with clinically significant flushing (121). E2 levels were also differently associated with the progression of atherosclerosis according to the time of initiation of hormonal replacement therapy (HRT). Higher levels of E2 decreased the rate of progression of carotid intima-media thickness in early postmenopausal women, but increased it in women in late menopause. These facts also support the “Timing Hypothesis” of HRT on reducing the atherosclerosis (3).”

120) Manson, J. E., Aragaki, A. K., Bassuk, S. S., Chlebowski, R. T., Anderson, G. L., Rossouw, J. E., … Prentice, R. L. (2019). Menopausal Estrogen-Alone Therapy and Health Outcomes in Women With and Without Bilateral Oophorectomy. Annals of Internal Medicine, 171(6), 406. https://doi.org/10.7326/M19-0274

121) Huang, A. J., Sawaya, G. F., Vittinghoff, E., Lin, F., & Grady, D. (2018). Hot flushes, coronary heart disease, and hormone therapy in postmenopausal women. Menopause, 25(11), 1286–1290. https://doi.org/10.1097/GME.0000000000001230

122) Clarkson, T.B.; Meléndez, G.C.; Appt, S.E. Timing hypothesis for postmenopausal hormone therapy. Menopause J. North Am. Menopause Soc. 2013, 20, 342–353, doi:10.1097/GME.0b013e3182843aad.

123 )Sriprasert, I., Hodis, H. N., Karim, R., Stanczyk, F. Z., Shoupe, D., Henderson, V. W., & Mack, W. J. (2019). Differential Effect of Plasma Estradiol on Subclinical Atherosclerosis Progression in Early vs Late Postmenopause. The Journal of Clinical Endocrinology & Metabolism, 104(2), 293–300. https://doi.org/10.1210/jc.2018-01600

Line 434:

However, oral administration of high dose of estrogen, associated or not with progestogen, may increase the risk of hypertension in aged postmenopausal women and different estrogen formulations may offer lower risks (128). Spontaneous menopause can induce metabolic syndrome and insulin resistance but is not associated with an increased risk of diabetes in advancing age. HRT may result in neutral or beneficial effects on glucose levels in patients with pre-existing type 2 diabetes, although oral treatment has superior performance than transdermal estrogen (129).

128)Swica, Y., Warren, M. P., Manson, J. E., Aragaki, A. K., Bassuk, S. S., Shimbo, D., … Womack, C. R. (2018). Effects of oral conjugated equine estrogens with or without medroxyprogesterone acetate on incident hypertension in the Women’s Health Initiative hormone therapy trials. Menopause, 25(7), 753–761. https://doi.org/10.1097/GME.0000000000001067

129)Cobin, R. H., & Goodman, N. F. (2017). American association of clinical endocrinologists and American college of endocrinology position statement on menopause - 2017 update. Endocrine Practice, 23(7), 869–880. https://doi.org/10.4158/EP171828.PS

Comment 4: I think the figure about genomic and nongenomic is quite confusing and not linked to the relevant content.

Response: The figure aims to summarize the contribution of selective agonists and antagonists for each estrogen receptor, while highlighting the signaling pathways involved in estrogen receptor-mediated effects. Additionally, the different responses elicited after nongenomic (through protein kinases) and gene transcription. This figure was repositioned to the end of session 3 (General Mechanism of Action of Estrogen Receptors). The following paragraph summarizes the content of figure.

Line 120:

In summary, canonical signaling through ERα is mediated by nuclear translocation and activation of gene transcription of anti-oxidative genes, like superoxide dismutase 2 (SOD2) on cellular stress conditions, as well as downregulation of inflammatory proteins, as tumor necrosis factor (TNF)-α. Similarly, ERβ induces upregulation of proteins involved in vascular tone control (NOS3 and PTGS2) and also downregulates proinflammatory and profibrotic gene programs. Membrane ERα and ERβ promote rapid increase of NO, by eNOS phosphorylation, and RhoA-associated kinase (ROCK) inhibition, and membrane ERβ also increases protein S-nitrosylation (SNO) via eNOS activation. Furthermore, GPER increases NO production, by eNOS phosphorylation, and increases ERK/ glycogen synthase kinase (GSK)-3β pathway activation while reduces TNF-α production via NFκB and decreases cyclin B1 and CDK1 activity. Therefore, the activation of these ER in the cardiovascular system could be explored to reduce cardiac remodeling and the progression of heart failure induced by MI.

Reviewer 2 Report

In their manuscript entitled, ‘Estrogen Receptors: Therapeutic Perspectives for the Treatment of Cardiac Dysfunction after Myocardial Infarction’, da Silva et al. are reviewing the current understanding underlying the molecular, preclinical, and clinical aspects of cardiovascular 17β-estradiol effects and estrogen receptor modulation as a potential therapeutic target for the treatment of myocardial infarction-induced cardiac dysfunction. Overall, the manuscript addresses an important question regarding the treatment of cardiac dysfunction after myocardial infarction. The review is accurate and the content is educative and well-integrated. However, the flow of the manuscript could be significantly improved, as in its current form, it can be challenging to follow. Below are some comments I have:

Major comments:

  1. The animal model and sex of some of the studies are missing in the text. It would be beneficial to have the findings of these studies highlighted in the text so that the reader can easily tell whether the data was from male or female animals. Thus, I suggest the authors to go through the entire manuscript and include this pivotal information.
  2. Throughout the manuscript, there are numerous run-on sentences which makes it challenging to follow at times. I would suggest cutting those sentences. Some examples include: lines 79 through 83, 119 through 122, lines 212 through 216, lines 355 through 358, etc.
  3. It is possible for the reader to get lost among the well-cited but numerous details. Would it be possible to summarize the essence of findings and potential clinical implications in a short paragraph at the end of each section?

Minor comments:

  1. Please spell out 17β-estradiol in the abstract.
  2. In line 57, please correct the citation.
  3. Please correct the wording in line 79, i.e “results promotes”
  4. In lines 101-104, please reword the sentence as the logic is confusing to follow for the reader.
  5. Line 102 should read “eNOS”.
  6. The abbreviations should be replaced from line 104 through 112 elsewhere.
  7. Line 155, I recommend changing “submitted” to “subjected”.
  8. Please correct for line 186, “Thus, it seems that the influence of ER on post-MI angiogenesis seems…”
  9. I suggest the authors to read the manuscript again and correct for consistency in terms of abbreviations. For example, although myocardial infarction is referred to as MI throughout the text, line 156 refers to it as myocardial infarction.
  10. Line 188, please delete “as”.
  11. Line 227, change “mediates” to “mediating”
  12. The sentence correlated to lines 251-253 is very difficult to follow, please consider rewording.
  13. Line 255, missing the word “of” before “gender or age”.
  14. When referring to Bopassa’s paper in line 309, it is not clear what the authors are trying to state regarding the perfusion during ischemia/reperfusion. It would be useful to revise this sentence and provide more details.

Author Response

Review Report 2

Comments and Suggestions for Authors

In their manuscript entitled, ‘Estrogen Receptors: Therapeutic Perspectives for the Treatment of Cardiac Dysfunction after Myocardial Infarction’, da Silva et al. are reviewing the current understanding underlying the molecular, preclinical, and clinical aspects of cardiovascular 17β-estradiol effects and estrogen receptor modulation as a potential therapeutic target for the treatment of myocardial infarction-induced cardiac dysfunction. Overall, the manuscript addresses an important question regarding the treatment of cardiac dysfunction after myocardial infarction. The review is accurate and the content is educative and well-integrated. However, the flow of the manuscript could be significantly improved, as in its current form, it can be challenging to follow. Below are some comments I have:

Major comments:

  1. The animal model and sex of some of the studies are missing in the text. It would be beneficial to have the findings of these studies highlighted in the text so that the reader can easily tell whether the data was from male or female animals. Thus, I suggest the authors to go through the entire manuscript and include this pivotal information.

Response: As suggested by the reviewer, authors described in the text models and sex of animals that was considered missing.

  1. Throughout the manuscript, there are numerous run-on sentences which makes it challenging to follow at times. I would suggest cutting those sentences. Some examples include: lines 79 through 83, 119 through 122, lines 212 through 216, lines 355 through 358, etc.

Response: The manuscript was revised and the sentences were modified as suggested.

  1. It is possible for the reader to get lost among the well-cited but numerous details. Would it be possible to summarize the essence of findings and potential clinical implications in a short paragraph at the end of each section?

Response: As suggested, a summary of the results and implications were included at the end of each session.

Line 85:

  1. Estrogen receptors and distribution in the cardiovascular system

Therefore, as the different ER subtypes show wide distribution in most cell types of the cardiovascular system, their actions have been extensively investigated for the treatment of CVD, including MI, which will be further explored in the following sessions.

Line 120:

  1. General mechanism of action of ER

In summary, activation of ERα promotes nuclear translocation and anti-oxidative genes transcription, superoxide dismutase 2 (SOD2), as well as downregulation of inflammatory proteins, tumor necrosis factor (TNF)-α. Similarly, ERβ induces upregulation of proteins involved in vascular tone control (NOS3 and PTGS2) and downregulates proinflammatory and profibrotic gene programs. Membrane ERα and ERβ promote rapid increase of NO, by eNOS phosphorylation, and RhoA-associated kinase (ROCK) inhibition, and membrane ERβ increases protein S-nitrosylation (SNO) via eNOS activation. Furthermore, GPER increases NO production, by eNOS phosphorylation, and increases ERK/ glycogen synthase kinase (GSK)-3β pathway activation while reduces TNF-α production and cyclin B1 and CDK1 activity. Therefore, the activation of ER in the cardiovascular system could be expected to reduce cardiac remodeling and the progression of HF induced by MI.

Line 346:

  1. ER modulation and MI: preclinical studies

E2 displays a broad range of cardioproctective effects and can be useful for treatment of CVD, especially MI. Besides anti-dyslipidemic, anti-atherogenic and vasodilator effects, E2 may exert direct cardiac effects, such as increasing cell survival, stimulating ischemic preconditioning, regulating myocardial hypertrophy and fibrosis, as well as normalizing cardiac function and rhythm. Activation of ER plays an important role in most E2 effects, with an interplay of ER subtype-selective actions generally seen in both sex. Among them, ERα stimulates myocardial regeneration and angiogenesis through activation of c-kit+ cardiac cells and endothelial progenitor cells (EPC), and improves antioxidant mechanisms and cardiac function while reduces vascular smooth muscle cells calcification, coronary tonus, fibrosis and post-MI mortality. In general, ERβ stimulation leads to upregulation of antioxidant and anti-apoptotic gene programs, although its role on attenuating fibrosis and arrhythmias has already shown. Besides, GPER activation not only generates coronary vasodilation and controls cholesterol catabolism and inflammation but also, improves cardiac conditioning after I/R injury. ERα reduces infarct size and ventricular remodeling but these beneficial actions prompt further research on estrogen effects in MI patients, as subsequently discussed.

Line 475:

  1. ER modulation and MI: clinical studies

Thus, as evidenced by most recent data, the benefits of HRT in reducing the risk of CVD in postmenopausal women seem to be related to the moment of therapy onset, being evident in the initial phase of menopause. In addition, the route of administration may also influence the outcomes and, while transdermal E2 patches are less likely to induce thrombotic events, oral E2 formulation provides an improvement in serum lipid profile and risk of atherosclerosis. It was also shown that SERM reduce the rate of infarction and hospitalizations for heart disease and may also improve lipid profile in women at increased risk of CVD. In this way, women undertaking HRT or using SERM could benefit from an improved quality of life and lower risks of CVD.

Line 519:

  1. Genetic factors related to estrogen receptors and MI

ER gene variations are important factors associated to risk for CVD in a sex-dependent manner, and may possibly determine pharmacological responses for ER-targeted therapies. While ERS1 variations in women shows correlation to increased risk for CVD including MI, ERS2 polymorphisms may increase MI susceptibility in men. However, ER gene variants may also predispose to increased risk factors for CVD, as seen in women bearing ERS2 variants or a hypo-functional GPER variant.

Minor comments:

  1. Please spell out 17β-estradiol in the abstract.

Response: The term has been added to the line 16.

  1. In line 57, please correct the citation.

Response:  The citation has been corrected, line 64

  1. Please correct the wording in line 79, i.e “results promotes”

Response:  The wording has been corrected, line 89

  1. In lines 101-104, please reword the sentence as the logic is confusing to follow for the reader.

Response: This sentence was part of the figure legend and was withdrawn from the revised manuscript

.Line 102 should read “eNOS”.

Response:  This has been corrected (figure legend)

  1. The abbreviations should be replaced from line 104 through 112 elsewhere.

Response: This sentence was part of the figure legend and was withdrawn from the revised manuscript

.Line 155, I recommend changing “submitted” to “subjected”.

Response:  This has been corrected, line 163.

  1. Please correct for line 186, “Thus, it seems that the influence of ER on post-MI angiogenesis seems…”x

Response:  For better understanding and fluidity of the text, we think it is better to remove this sentence. Lines 194-199.

  1. I suggest the authors to read the manuscript again and correct for consistency in terms of abbreviations. For example, although myocardial infarction is referred to as MI throughout the text, line 156 refers to it as myocardial infarction.

Response: The manuscript was revised to correct for consistency in terms of abbreviations

  1. Line 188, please delete “as”.

Response: This has been corrected, line 194

  1. Line 227, change “mediates” to “mediating”

Response: This has been corrected, line 229.

  1. The sentence correlated to lines 251-253 is very difficult to follow, please consider rewording.

Response:  The sentence has been corrected, lines 254-256

  1. Line 255, missing the word “of” before “gender or age”.

Response: This has been corrected, line 258.

  1. When referring to Bopassa’s paper in line 309, it is not clear what the authors are trying to state regarding the perfusion during ischemia/reperfusion. It would be useful to revise this sentence and provide more details.

Response: To improve the text has been rewritten, lines 300-302

Round 2

Reviewer 1 Report

I'm fine with the revised manuscript.